# Diffusion Models-Based Purification for Common Corruptions on Robust 3D Object Detection

**DOI:** 10.3390/s24165440

**Published:** 2024-08-22

**Authors:** Mumuxin Cai, Xupeng Wang, Ferdous Sohel, Hang Lei

**Affiliations:** 1School of Information and Software Engineering, The University of Electronic Science and Technology of China, Chengdu 610054, China; caimumuxin@std.uestc.edu.cn (M.C.);; 2Laboratory Of Intelligent Collaborative Computing, The University of Electronic Science and Technology of China, Chengdu 610054, China; 3School of Information Technology, Murdoch University, Perth, WA 6150, Australia; f.sohel@murdoch.edu.au

**Keywords:** 3D object detection, LiDAR scene data, point cloud, diffusion models, defence strategy, adversarial robustness

## Abstract

LiDAR sensors have been shown to generate data with various common corruptions, which seriously affect their applications in 3D vision tasks, particularly object detection. At the same time, it has been demonstrated that traditional defense strategies, including adversarial training, are prone to suffering from gradient confusion during training. Moreover, they can only improve their robustness against specific types of data corruption. In this work, we propose LiDARPure, which leverages the powerful generation ability of diffusion models to purify corruption in the LiDAR scene data. By dividing the entire scene into voxels to facilitate the processes of diffusion and reverse diffusion, LiDARPure overcomes challenges induced from adversarial training, such as sparse point clouds in large-scale LiDAR data and gradient confusion. In addition, we utilize the latent geometric features of a scene as a condition to assist the generation of diffusion models. Detailed experiments show that LiDARPure can effectively purify 19 common types of LiDAR data corruption. Further evaluation results demonstrate that it can improve the average precision of 3D object detectors to an extent of 20% in the face of data corruption, much higher than existing defence strategies.

## 1. Introduction

With the rapid development of Light Detection and Ranging (LiDAR) sensors, their ability to capture large-scale point cloud data has been extensively utilized. LiDAR sensors are advanced remote sensing devices that utilize laser technology to precisely measure distances and acquire detailed 3D spatial information. The process begins with the emission of laser pulses from the sensor, which are then directed towards the target area. Upon striking an object, the pulses reflect back and are captured using the sensor’s receiver. By measuring the time it takes for the laser pulses to travel to and from the object, combined with the precise angular measurements of the laser’s direction, the LiDAR sensor calculates the object’s 3D position. This process is repeated rapidly, scanning across the entire target area, generating a dense point cloud of millions of 3D points that accurately represent the environment’s geometry and topology. The resulting data are then processed and analyzed to extract meaningful information for applications such as topographic mapping, urban planning, environmental monitoring, and autonomous navigation systems. In particular, LiDAR-based 3D object detection models have found wide applications in various fields, such as autonomous driving [1]. However, these neural network-based deep models are susceptible to adversarial attacks that can easily deceive models and make incorrect predictions [2,3]. Notably, 3D object detection models have also been proven to be very vulnerable to adversarial attacks [4,5]. The data collected using LiDAR sensors present some common corruptions, such as noise and low density [6]. Possible data corruption can be exploited as adversarial attacks, posing a significant threat to security critical applications. Therefore, there is a need for innovative techniques to purify corrupted data and enhance the robustness of 3D object detection-based systems.

There are two main types of defense strategies for adversarial attacks, namely adversarial training [7,8] and adversarial purification [9,10]. Adversarial training utilizes adversarial samples to train neural network models, making them robust against the same type of adversarial attacks. This is a commonly used standard defense strategy, as it is very effective in dealing with fixed types of adversarial attacks and can directly improve the robustness of the model. Adversarial training is usually aimed at defending against specific adversarial attacks, which leads to a lack of flexibility and difficulty in coping with multiple types of simultaneous adversarial attacks. Deep models that have undergone specific adversarial training are often very fragile and inflexible, when faced with other types of adversarial attacks. This requires exhausting adversarial training to compensate. In addition, typical adversarial training techniques suffer from a much higher training cost and computational complexity in contrast to standard training methods. On the other hand, adversarial purification relies on generative models to purify input adversarial samples, making them harmless and unable to interfere with deep models. Compared to adversarial training, adversarial purification is more flexible and can handle a wider range of adversarial threats, even performing well in the face of unknown adversarial attacks. In addition, adversarial purification is typically trained and operated independently of task models, while adversarial training must bound to neural networks for training together. Adversarial training can be seen as an improvement to the detector, while adversarial purification is not a problem that should be considered for detectors. The detector improvement and data purification are two separate research areas. Data purification focuses on purifying adversarial samples, whereas the goal of a detector is to predict normal samples as correctly as possible. If the detector itself considers data purification, it will inevitably affect its performance on normal samples. However, the existing adversarial purification method is limited by the ability of generative models, and often lags behind adversarial training in improving robustness performance [11]. The capabilities of the generative models used, such as generative adversarial networks (GANs) [12] and variational autoencoder (VAE) [13], cannot meet the needs of adversarial purification, resulting in their limited robustness performance.

The technique of diffusion models has emerged with its powerful generative ability surpassing GANs and VAE [14,15]. In addition to the 2D images, diffusion models have also demonstrated superior performance in the field of 3D point cloud generation [16] and purification [17]. Therefore, we consider utilizing the technique in large-scale LiDAR scenes to purify corrupted samples. The diffusion model mainly consists of forward diffusion process and reverse diffusion process. The forward diffusion process follows a Markov chain, gradually adding noise to the input sample. When there are enough steps, the input sample can be transformed into an ideal noise model. The reverse diffusion process, on the other hand, involves continuously removing noise from noisy images, which is usually conducted using neural networks and ultimately restores the noise to a clear original image. The diffusion model’s ability to recover from noise introduces significant randomness. This makes it more flexible when used in adversarial purification defense strategies and it is able to cope with a wider variety of unknown adversarial attacks.

In this paper, we propose an adversarial purification method to purify common corruptions in large-scale LiDAR scene data, named LiDARPure. The proposed purification method is detector independent and can be equipped to any detector. It can be used for low frame rate 3D object detection applications with existing hardware, such as robotics. With the improvement of computational power, it can also be applied to higher real-time 3D object detection applications in the future. Compared with adversarial training, LiDARPure leverages the generative ability of diffusion models, enabling it to handle various types of data corruption without being hindered by data sparsity and gradient confusion. The corrupted LiDAR scene data are transformed into random noise through the forward diffusion process of the diffusion model, in which a large amount of randomness is introduced to smooth out the corruption in the data. Subsequently, the noise is restored to a purified LiDAR scene using the reverse diffusion process. It should be noted that we extract latent representations in units of voxels and use them as conditions to help the denoising in the reverse diffusion process. In addition, we use the proposed denoising recovery loss to supervise the reverse diffusion process, making the purified sample closer to the pure data.

The main contributions of this article can be summarized as follows:We demonstrate that existing standard adversarial training has significant limitations for 3D object detection models because of the large-scale space, sparsity, variety of corruptions, and gradient confusion.We propose LiDARPure, which leverages the powerful generative ability of diffusion models to purify common corruptions on LiDAR scenes and works independently of the 3D object detector.We propose a denoising recovery loss term, which utilizes the latent features and geometry information to supervise the reverse diffusion process.The experiments demonstrate that the proposed method achieves superior robustness to corruptions and outperforms existing 3D point cloud purification methods.

## 2. Related Work

### 2.1. LiDAR-Based 3D Object Detection

In recent years, 3D object detection with LiDAR sensors has witnessed significant advancements, enabling practical perception in autonomous driving and robotics. Notably, VoxelNet [18] pioneered the use of voxel feature encoding to effectively represent point clouds, setting the foundation for subsequent works. SECOND [19] adopted a sparse 3D convolutional network to process the voxelized point clouds, significantly reducing computational complexity and achieving faster inference speeds while maintaining high detection accuracy. PointPillars [20] projected point clouds onto a 2D grid in a bird’s eye view, forming pillars, and then processed these pillars using 2D convolutions. It further simplified the representation by encoding point clouds into pillars, striking a balance between performance and efficiency. PointRCNN [21] introduced a two-stage approach, which first generated 3D proposals from raw point clouds and then refined them, enhancing the recall rate of point cloud segmentation and detection accuracy simultaneously. PV-RCNN [22] extended PointRCNN by incorporating voxel-based features into its refinement stage, demonstrating the effectiveness of combining voxel-based efficiency with point-based accuracy. CenterPoint [23] and CenterFormer [24] adopted a center-based paradigm, which directly predicted object centers and their attributes, and demonstrated state-of-the-art results, simplifying the detection pipeline. These models have significantly contributed to the field, advancing the capabilities of 3D object detection from LiDAR data.

### 2.2. Adversarial Attack against 3D Object Detection

LiDAR sensors are widely used in 3D vision tasks, but the data they collect poses many challenges, such as large-scale range and sparsity. The existing adversarial attacks [4,5,25,26] have proven that 3D object detection is highly vulnerable to malicious samples. Cao et al. [25] conducted the first security study of LiDAR-based perception in autonomous vehicles, exploring LiDAR spoofing attacks by adding malicious vehicle point clouds to areas obscured by vehicles, achieving attack success rates of around 75%. Tu et al. [4] deceived LiDAR detectors by placing the adversarial mesh on vehicle rooftop, which is disguised as cargo carried by vehicles. Wang et al. [5] used optimized adversarial perturbations to attack the entire scene, significantly reducing the detector’s performance to detect all targets. The points collected using LiDAR often suffer from data corruption, which can lead to instability and pose significant safety challenges for 3D object detection tasks. Shin et al. [27] proved that additional light may cause LiDAR to make incorrect distance judgments on objects. LISA [28] has demonstrated through simulation that adverse weather is also an important cause of data corruption in LiDAR. Heavy fog [29,30] and snowfall [31] have been shown to cause a significant amount of noise and point shift in the scene. In addition to data corruption caused by adverse weather, LiDAR may also spontaneously have local and global noise, density degradation, shearing, and object rotation [32]. Dong et al. [6] summarized 27 common data corruptions in autonomous driving scenarios and tested the robustness of mainstream detectors. Among them, 19 types of data corruption completely occur on LiDAR sensors. These data corruption types are different, making it difficult to completely eliminate through adversarial training. This indicates the need for more universal and flexible methods to eliminate the impact of LiDAR data corruption on 3D object detection models.

### 2.3. Defend Strategies for 3D Computer Vision Methods

Defense strategies tailored for 3D computer vision tasks have attracted attentions from the research community with the extensive studies on their adversarial counterparts. Generally, existing defense methods can be divided into to adversarial training and purification. Hahner et al. [29,31] simulated the data captured in adverse weather and used them for adversarial training, which greatly enhances the robustness of 3D object detection models to heavy fog and snowfall weather. Teufel et al. [33] used data simulation and data augmentation methods to train a 3D object detector in three types of severe weather conditions, enabling it to have high adversarial robustness to rain, snow, and fog weather simultaneously. 3D-Vfield [26] improves the performance of the model by adding possible unknown objects to the scene data trained in the model. However, these methods of adversarial training can only be effective for specific types of attacks. In 3D recognition tasks, DUP-Net [34] purified the samples by denoising and upsampling, removing the outliers and noise points from adversarial samples. Dong et al. [35] filtered the vector directions of various regions in the point cloud and fed the purified features into the classifier to avoid adversarial attacks. SUN et al. [36] designed a self supervised mechanism for 3D recognition models, which enhanced the robustness of the model during training. IF-Defense [37] and LPC [38] purify adversarial samples through implicit functions and gradient rules. Currently, adversarial purification is mainly used for defense against point cloud object recognition, while adversarial training is usually used for large-scale LiDAR scenes. The 3D object detection model requires more efficient adversarial purification methods to cope with the corruption of large-scale LiDAR data.

### 2.4. Diffusion Models for 3D Data

Both discrete-time diffusion models [14] and continuous time diffusion models [15] have shown their potential in the field of two-dimensional image generation, and some studies [16,39,40,41] have also begun to use diffusion models to generate and complete three-dimensional point clouds. Lee et al. [42] and Nunes et al. [43] also demonstrated the ability of diffusion models to generate and complete large-scale real-world scenes. In addition, DiffPure [44] demonstrates the adversarial purification ability of diffusion models when used in defense strategies in the field of images. Subsequently, the diffusion model was also introduced into the adversarial purification of point clouds. PointDP [17] used the purification method derived from diffusion models to improve the robust precision of 3D recognition models, surpassing existing advanced IF-Defense [37] and LPC [38] methods. This proves the effectiveness of diffusion models in combating adversarial attacks on 3D data. The above research indicates that using diffusion models and adversarial purification to process large-scale LiDAR scene data and their common data corruptions is promising. However, these defense strategies based on diffusion models for 3D data are only applicable to dense point clouds of individual objects, focusing on 3D recognition models, which cannot be well applied to large-scale LiDAR scenes. The LiDARPure proposed in this paper can handle sparser large-scale spaces and purify corrupted scenes simultaneously.

## 3. Limitations of Adversarial Training on 3D Object Detection

Adversarial training, as a training method used in machine learning and deep learning to improve model robustness and generalization ability, has been widely used in the field of 2D images for a long time. However, in the task of 3D point cloud processing, its performance always has certain limitations. As Sun et al. [17] found, in 3D point cloud recognition tasks, the gradient of the loss function is difficult to backpropagate to the input data during adversarial training due to a large number of *k*NN layers, top-*k* operations, and max pooling operations. There is a serious gradient confusion among them, which can lead to non-differentiability in some cases and greatly increases the difficulty of training. Compared to the fixed resolution of 2D images, the disorder and arbitrary number of points in point cloud data, as well as the sampling algorithm used, will increase a lot of randomness during the training process. The use of sampling algorithms is reasonable for standard training, but the randomness introduced will greatly increase the workload and computational cost of adversarial training.

In addition to the limitations of adversarial training in the 3D point cloud recognition tasks mentioned above, we focus more on the performance of adversarial training in large-scale LiDAR data and 3D object detection models. The number of points in large-scale LiDAR point cloud scenes is usually much larger than that of a single point cloud object. Three-dimensional object detection models will lose more information and introduce more randomness during data sampling. Compared to 3D point cloud recognition models, the tasks completed using 3D object detection models are more complex, requiring simultaneous localization and a classification of objects. This increases the complexity of its deep neural network model and may use more top-*k* and max pooling operations that can cause gradient confusion. In fact, a large number of 3D object detection models use PointNet [45] or PointNet++ [46] as part of their feature extraction structures, such as PointPillars [20], PointRCNN [21], and CenterPoint [23]. In adversarial attack work [5], we have found that the Non-Maximum Suppression (NMS) operation introduced by existing 3D object detection models in object localization also introduces significant randomness. This hinders not only the optimization of the gradient against adversarial attacks, but also the fitting against adversarial training.

In addition to the challenges posed by the increase in data size and network complexity for adversarial training, 3D object detection faces more possible data corruptions compared to point cloud recognition tasks. The main data corruptions that may be encountered in point cloud recognition tasks is local density reduction and noises. In the recognition task, a single object point cloud does not need to consider the impact of adverse weather such as heavy fog and snowfall, which will add a large number of additional noise points to the scene. In addition, LiDAR point cloud scenes are also susceptible to unique data corruption such as shearing and spatiotemporal misalignment. Adversarial training usually only deals with specific adversarial attacks or data corruption; Hahner et al. [29,31] demonstrated that simulating and adversarial training against adverse weather can improve the robustness of 3D object detection models. However, it is unreasonable to conduct adversarial training on data corruption in all common types of LiDAR scenarios. Firstly, Dong et al. [6] proposed 27 common data corruptions issue faced by LiDAR point cloud data. There are so many types of them that adversarial training requires significant computational cost, and deep networks need to consider all the corruptions simultaneously, making it difficult to fit. Secondly, adversarial training is not a perfect solution. Lechner et al. [47] demonstrated that adversarial training targeting specific adversarial attacks may reduce the performance of the models and make it more vulnerable to other types of adversarial attacks. We selected three adverse weather corruptions simulation methods in LiDAR scenarios, namely snow [31], rain [28], and fog [29], and studied their impact with related adversarial training on the KITTI dataset. The three simulation methods for adverse weather conditions generated corrupted samples on the training set of the KITTI dataset were used for the adversarial training of the PointPillars detector. The malicious samples simulated using adverse weather on the KITTI validation set are used to evaluate the robustness of adversarial training against an unknown corruption. In Table 1, the performance of the PointPillars detector is demonstrated in the form of AP when it has undergone different adversarial training and encountered data corruptions, respectively.

The results presented in Table 1 indicate that adversarial training can improve the robustness of detectors against the same type of data corruption. But at the same time, adversarial training may reduce the standard performance of 3D object detection models and make them more vulnerable to other data corruptions. This also proves that for LiDAR point cloud data with a large number of common types of corruption, it is not applicable to use adversarial training to deal with all data corruptions.

## 4. LiDARPure: Purification for LiDAR Scenes

In this section, we first present the fundamentals of the diffusion models and their applications in 3D point cloud data. Next, the problem studied in this paper is formulated in Section 4.2. We describe the proposed LiDARPure method for purifying LiDAR scenes in detail in Section 4.3. The loss functions used for the training of the diffusion model-based purification are then described in Section 4.4.

### 4.1. Preliminaries

Diffusion models can be classified into discrete diffusion models and continuous diffusion models. In this paper, we mainly utilize the discrete diffusion model and one of its solutions, Denoising Diffusion Probabilistic Models (DDPM) [14]. According to DDPM, the discrete-time conditional diffusion model can be formulated as follows:

Given a sample $x_{0}∼q\left(x\right)$ sampled from the data distribution q(x), the forward diffusion process is defined as a Markov chain that gradually adds Gaussian noise ϵ to the sample over *T* steps, thus generating a series of noisy samples $X=\{x_{1},\ldots ,x_{T}\}$. The noise added in each step is controlled by forward variances $\beta =\left\{\beta_{1},\ldots ,\beta_{T}\right\}\in \left(0,1\right)$. Thus, $x_{t}$ can be represented as follows:(1)$$x_{t}=\sqrt{1-\beta_{t}}x_{t-1}+\sqrt{\beta_{t}}\epsilon_{t-1},\;\epsilon_{t-1}∼N\left(0,I\right),$$

The conditional distribution of $q(x_{t})$ can then be represented as follows:(2)$$q\left(x_{t}|x_{t-1}\right)=N\left(x_{t};\sqrt{1-\beta_{t}}x_{t-1},\beta_{t}I\right),$$
(3)$$q\left(x_{1:T}|x_{0}\right)=\prod_{t=1}^{T}q(x_{t}|x_{t-1}).$$

Given $\alpha_{t}=1-\beta_{t}$, $x_{t}$ and reverse diffusion process can be written as follows:(4)$$x_{t}=\sqrt{{\bar{\alpha }}_{t}}x_{0}+\sqrt{1-{\bar{\alpha }}_{t}}\epsilon ,$$
(5)$$q\left(x_{t}|x_{0}\right)=N\left(x_{t};\sqrt{{\bar{\alpha }}_{t}}x_{0},\left(1-{\bar{\alpha }}_{t}\right)I\right).$$

During the diffusion process, the information of the data $x_{0}$ is gradually destroyed. As T→∞, the entire scene is imposed with added Gaussian noise and $q\left(x_{T}\right)\approx N\left(0,I\right)$.

The reverse denoising diffusion process aims at solving the inverse process $q(x_{t-1}|x_{t})$ for the forward diffusion process $q(x_{t}|x_{t-1})$. An existing study [15] on the theory of stochastic differential equations proves that $q(x_{t-1}|x_{t})$ also approximately obeys a Gaussian distribution when $\beta_{t}$ is sufficiently small. Thus, it is only necessary to obtain the noise $\epsilon_{\theta }\left(x_{t},t\right)$ at step *t* to calculate $x_{t-1}$. The reverse diffusion process is difficult to compute directly, so the DDPM uses the neural network $p_{\theta }\left(x_{t-1}|x_{t}\right)$ to fit the $q(x_{t-1}|x_{t})$. The goal is to make the predicted noise and the actual added noise as identical as possible. The conditional probability distribution of the neural network $p_{\theta }\left(x_{t-1}|x_{t}\right)$ can be represented as follows:(6)$$p_{\theta }\left(x_{t-1}|x_{t}\right)=N\left(x_{t-1};\mu_{\theta }\left(x_{t},t\right),\Sigma_{\theta }\left(x_{t},t\right)\right),$$
(7)$$p_{\theta }\left(x_{0:T}\right)=p\left(x_{T}\right)\prod_{t=1}^{T}p_{\theta }\left(x_{t-1}|x_{t}\right).$$

Although the probability distribution $q(x_{t-1}|x_{t})$ of the reverse diffusion process is difficult to solve, one can derive $q(x_{t-1}|x_{t},x_{0})$, given the condition $x_{0}$:(8)$$q\left(x_{t-1}|x_{t},x_{0}\right)=N\left(x_{t-1};{\tilde{\mu }}_{\theta }\left(x_{t},x_{0}\right),{\tilde{\beta }}_{t}I\right).$$

With the variance and mean above, the analytic form of $q(x_{t-1}|x_{t},x_{0})$ is obtained. The objective function of the training diffusion models is to minimize the negative log-likelihood of $p_{\theta }\left(x_{0}\right)$ by constructing a variational bound.

In the reverse diffusion process, additional information about the data can be input to assist data generation, which is called the conditional diffusion model. In the diffusion model-based 3D point cloud generation methods, due to the lack of texture information and the sparsity of point cloud data, the latent geometric feature *z* of point clouds are often used as conditions in the reverse diffusion process to help restore the original point cloud [16]:(9)$$p_{\theta }\left(x_{0:T}|z\right)=p\left(x_{T}\right)\prod_{t=1}^{T}p_{\theta }\left(x_{t-1}|x_{t},z\right),$$
where *z* is the latent feature extracted using encoder *E* from point cloud data *x*. Usually, the encoder *E* and noise predictor $\epsilon_{\theta }\left(x_{t},t,z\right)$ are trained together. The diffusion model has demonstrated a strong generation capability on point cloud data, and also has great potential in large-scale LiDAR scenes, making it possible to purify common corruptions in LiDAR scenes.

### 4.2. Problem Formulation

Given a real-world scenario *P*, a LiDAR sensor collects it and ideally obtains a pure LiDAR point cloud scene *S*. The scene *S* contains *m* objects $B=\{b_{1},\ldots ,b_{m}\}$ and their corresponding bounding boxes $b_{m}=\left(x,y,z,l,w,h,\theta \right)$, where (x,y,z) represents the center point coordinates of the objects, (l,w,h) represents the length, width, and height of the bounding box, and θ represents the orientation of the objects in the bird’s eye view. The goal of 3D object detector *D* is to output predictions that match the bounding boxes of the objects, which means D(S)=B. However, during the data collection process, due to sensor defects or adverse weather conditions, it often leads to data corruption such as noise *n*. A pure LiDAR scene *S* becomes a corrupted LiDAR scene $\hat{S}$ when these corruptions *n* are imposed, which can be written as $\hat{S}=S+n$. The corruption of data causes errors in the detection result $D(\hat{S})$ of the detector *D*, which is different from the detection result D(S) of pure LiDAR scenes, i.e., $D\left(\hat{S}\right)\neq D\left(S\right)$. The goal of this study is to purify the corrupted LiDAR scene *S* based on the powerful generation ability of diffusion models. By adding noise through the forward diffusion process of *T* steps, the corrupted scene is transformed into Gaussian noise $S^{T}$, and then gradually denoised through the reverse diffusion process to obtain the purified point cloud scene $S^{\prime }$, which needs to be as similar as the pure scene *S* as possible. At this point, detector *D* has the same prediction results for them, which means $D\left(S^{\prime }\right)=D\left(S\right)$.

### 4.3. Design of LiDARPure

As shown in Figure 1, the corrupted LiDAR scene data are progressively added with noise through a diffusion process to obtain a diffuse scene, and a reparameterized noise predictor is utilized in the denoising process of reverse diffusion to remove both the added noise and the corruption in the scene. Based on the LiDAR scene-scale DDPM in [43], we optimize the denoising process so that the local features at each point are involved in the noise predictor. In addition, we use a conditional diffusion model based on latent features to assist in the noise predictor.

We choose to use the conditional diffusion model to add noise and denoise from the scene point cloud, smoothing out corrupt information in the data. Different from single object points in 3D recognition tasks, large-scale LiDAR has a larger scene range and more sparse point cloud data. Therefore, referring to the 3D detector CenterPoint [23], we divided the LiDAR scene into several units using voxels in VoxelNet [18]. Additionally, utilizing pillars in PointPillars [20] instead of voxels is optional. If the corrupted scene $\hat{S}$ is divided into *N* units, it can be represented as $\hat{S}=\left(u^{1},\ldots ,u^{N}\right)$. Unlike the approach of Nunes et al. [43], we perform the diffusion process on a per unit basis. Therefore, for the reverse diffusion process, its output is $S^{\prime }=\left(u^{1^{\prime }},\ldots ,u^{N^{\prime }}\right)$. By using the latent feature extractor *E* for each unit in VoxelNet or PointPillars, the latent features of all units in the corrupted scene can be represented as $Z=\left(z^{1},\ldots ,z^{N}\right)=E\left(\hat{S}\right)$.

According to Formula (Equation 4), the forward diffusion process of unit $u^{n}$ can be described as follows:(10)$$u_{t}^{n}=\sqrt{{\bar{\alpha }}_{t}}u_{0}^{n}+\sqrt{1-{\bar{\alpha }}_{t}}\epsilon ,\;\forall u_{n}\in \hat{S},$$
where ϵ∼N(0,I) and ${\bar{\alpha }}_{t}=\prod_{i=1}^{t}\left(1-\beta_{i}\right)$. During the forward diffusion process, the points within each unit and the entire LiDAR scene gradually transform into Gaussian noise. The corrupt information is gradually smoothed out, and the geometric information of the scene is also blurred. Therefore, in the process of reverse diffusion, we utilize the latent features of each unit to assist the generation and restoration of the scene. We calculate the reverse diffusion process for the *n*-th unit using Formula (Equation 9), which can be rewritten as follows:(11)$$p_{\theta }\left(u_{0:T}^{n}|z^{n}\right)=p\left(u_{T}^{n}\right)\prod_{t=1}^{T}p_{\theta }\left(u_{t-1}^{n}|u_{t}^{n},z^{n}\right),$$
(12)$$p_{\theta }\left(u_{t-1}^{n}|u_{t}^{n},z^{n}\right)=N\left(u_{t-1}^{n}|\mu_{\theta }^{n}\left(u_{t}^{n},t,z^{n}\right),\beta^{n}I\right),$$
where $\mu_{\theta }^{n}\left(u_{t}^{n},t,z^{n}\right)$ represents the approximate mean value of neural network parameterization. With the introduction of conditional latent features $z_{n}$, $\mu_{\theta }^{n}\left(u_{t}^{n},t,z^{n}\right)$ can be further written according the reparameterization method similar to DDPM:(13)$$\mu_{\theta }^{n}\left(u_{t}^{n},t,z^{n}\right)=\frac{1}{\sqrt{1-\beta_{t}}}\left(u_{t}^{n}-\frac{\beta_{t}}{\sqrt{1-{\bar{\alpha }}_{t}}}\epsilon_{\theta }^{n}\left(u_{t}^{n},t,z^{n}\right)\right).$$Based on Formula (Equation 11), the reverse diffusion process of *n*-th unit in the scene can be approximated using a neural network, which can be expressed as follows in step *t*:(14)$$u_{t-1}^{n}=u_{t}^{n}-\frac{1-\alpha_{t}}{\sqrt{1-{\bar{\alpha }}_{t}}}\epsilon_{\theta }^{n}\left(u_{t}^{n},t,z^{n}\right)+\frac{1-{\bar{\alpha }}_{t-1}}{1-{\bar{\alpha }}_{t}}\beta_{t}N\left(0,I\right),$$
where $\epsilon_{\theta }^{n}\left(u_{t}^{n},t,z^{n}\right)$ is the noise predictor at time *t* about $u_{t}^{n}$. The noise predictor $\epsilon_{\theta }^{n}$ is combined with the encoder *E* in the unit for training to ensure that the diffusion model better utilizes the extracted latent features. We independently perform forward diffusion and reverse diffusion processes for each unit, and utilize their geometric latent features as conditions in the reverse denoising process. Based on the above formula and derivation, we can use the conditional diffusion model to purify each unit in the scene. It is important to note that the flexibility of LiDARPure allows it to select arbitrary spatial processing and point cloud feature extraction units. Voxels and pillars are optional large-scale scene feature extraction units that are completely independent, and the performance analysis on these two units is provided in the experiments section.

### 4.4. Loss Function

The noise predictor $\epsilon_{\theta }^{n}$ is trained to predict the noise at each step of the denoising process. Based on this noise predictor, the difference between the prediction and Gaussian noise ϵ∼N(0,I) can be calculated. For the *n*-th voxel or pillar unit, the noise difference at step *t* can be computed as the l2-norm distance:(15)$$dis^{n}=||\epsilon -\epsilon_{\theta }^{n}\left(u_{t}^{n},t,z^{n}\right){\left|\right|}_{2}.$$

The above distance measurement is used for each step of the reverse diffusion process of the independent unit, making the output of the noise predictor more accurate and restoring the most realistic unit during the denoising process. For the entire scene, the reverse diffusion process of all units is synchronous, so the single step denoising distance loss of the entire LiDAR scene is formulated as follows:(16)$$L_{dis}=\sum_{n=0}^{N}dis^{n}.$$

Considering that the latent features of the scene serve as conditions to supervise the entire reverse diffusion process and play an important role in 3D object detection, it is crucial to pay attention to the changes in latent features after the reverse diffusion process. Therefore, the latent feature recovery measurement is proposed to measure the latent feature differences between the scene purified using the diffusion process and the pure scene. Similar to the distance loss $L_{dis}$, we independently calculate the difference in the latent features of each voxel or pillar unit’s point cloud after the diffusion process:(17)$$diff^{n}=\left|\right|{\bar{z}}^{n}-z_{0}^{n}{\left|\right|}_{2},$$
where ${\bar{z}}^{n}$ represents the latent feature extracted using the *n*-th unit of the pure sample. $z_{0}^{n}$ represents the corresponding latent feature of the purified sample after the t-step reverse diffusion process. In addition to the differences between abstract latent features, we believe that maintaining the geometric position of the point clouds in each unit is equally important. We use the Hausdorff distance to control the geometric information of point clouds within each unit:(18)$$pos^{n}=Hausdorff\left({\bar{u}}^{n},u^{n^{\prime }}\right),$$
where $\bar{u}$ represents the point cloud units in the pure sample. The Hausdorff distance is often used to measure the maximum difference in geometric information between two point cloud sets.

The latent features and geometric information of each unit after purification are controlled based on $diff^{n}$ and $pos^{n}$. Therefore, the denoising recovery loss of the entire scene can be represented as follows:(19)$$L_{rec}=\sum_{n=0}^{N}\left(diff^{n}+pos^{n}\right).$$

The denoising recovery loss ensures that the latent features and geometric information of the purified scene after the reverse diffusion process are closer to the pure scene, which makes it perform better as the input in subsequent 3D object detection tasks.

## 5. Experiments

In this section, the experimental results of the proposed LiDARPure method are reported. First, we introduce our experimental setup in Section 5.1. The quantitative results and qualitative results are presented in Section 5.2 and Section 5.3. In addition, ablation studies are discussed in detail in Section 5.4.

### 5.1. Experimental Setup

**Datasets and Detectors.** In this part, we describe the experimental setup focusing on the dataset and the victim 3D object detectors employed for evaluation. KITTI [48], a seminal dataset for autonomous driving research, boasts a wealth of dedicated LiDAR scene data, comprising 3712 training samples and 3769 validation samples, among others. These samples encapsulate a diverse range of real-world driving scenarios, from bustling urban streets to open highways, ensuring a comprehensive evaluation of 3D object detection methods.

As for the 3D object detectors utilized in our experiments, we selected three prominent approaches: PointPillars [20], PointRCNN [21], and PV-RCNN [22]. PointPillars revolutionizes point cloud representation through pillars, achieving a balance between efficiency and accuracy. PointRCNN adopts a two-stage architecture, generating and refining 3D proposals, and is adept at handling sparse and noisy LiDAR data. PV-RCNN combines voxel-based and point-based features, enhancing global and local context understanding. Together, these detectors showcase the diversity and progress in 3D object detection from LiDAR point clouds.

**Common corruptions.** Our study builds upon the seminal work presented by Dong et al. [6], which introduces an extensive benchmark of 27 common data corruptions and their corresponding simulation methods, tailored to mimic the diverse challenges encountered by LiDAR sensors and 3D object detection. To address these corruptions, we utilized the proposed LiDARPure method to purify the damage in the data through a diffusion model. Our study employs the comprehensive corruption framework proposed in the aforementioned paper as a testbed and conducted detailed adversarial purification experiments on 19 types of LiDAR data corruption. Dong et al. divided each type of corruptions into five levels, and we chose level three with moderate corruption for experiments that are closer to real scenarios. The detection performance with LiDARPure purified scene data can demonstrate their effectiveness in enhancing the robustness of 3D object detection systems against these challenging conditions.

**Evaluation Metrics.** The commonly used evaluation metrics in 3D object detection are average precision and mean average precision. Considering the most common scenario, we use robust average precision to measure the performance of the proposed method, which represents the AP of defense strategies in responding to adversarial attacks. In addition, the first column in the table represents its standard AP without applying any defense strategy on pure data for comparison. We choose vehicles with moderate difficulty in the dataset to calculate robust AP. For each detector, the robust AP without LiDARPure applied for adversarial purification is also included to show the impact of LiDARPure on the performance.

**Implementation details.** During the training process, we selected the Adam [49] optimizer and set the learning rate to $1\times 10^{-4}$. A total of 80 epochs were trained, halving the learning rate every 20 epochs. For the parameters of the diffusion model, we referred to the method of Nunes et al. [43] and set $\beta_{0}$ and $\beta_{T}$ to $3.5\times 10^{-5}$ and $5\times 10^{-3}$, respectively. The number of forward diffusion steps *T* was set to 1000. In the process of reverse diffusion, we used DPMSolver [50] to accelerate denoising, which can reduce the denoising steps *T* to 50. When voxels were selected for feature extraction, their resolution was set to 0.2 m. For pillars, the resolution was 0.15 m. A more detailed discussion on resolution will be provided in ablation studies (Section 5.4). After the diffusion process, redundant scene points were sampled to the same number of points before being fed into the detector. The experiments were conducted on a personal computer equipped with an i7 13700 CPU, 64 GB of RAM, and an RTX 4090 graphics processing unit.

### 5.2. Comparison with State-of-the-Art Methods

In this section, we demonstrate LiDARPure’s adversarial purification performance against common data corruptions and compare it with state-of-the-art defense strategies. The methods of adversarial training mainly include LISA [28] and adversarial simulation by Hahner et al. [29,31]. In addition, Teufel et al.’s adversarial training based on data augmentation method [33] is also used for performance comparison. To our knowledge, there is currently no adversarial purification method specifically designed for LiDAR sensors and 3D object detection. Therefore, we chose the purification methods DUP-Net [34] and IF-Defense [37] in 3D point cloud recognition for comparison. LPC [38] is not chosen because its structural design can only be used for the defense of point cloud classification models, making it difficult to apply to 3D object detection models.

The average precision of the PointPillars detector under data corruptions with defense strategy is shown in Table 2. The voxel and pillar in bracket denote the units utilised to perform the potential feature extraction and diffusion processes, respectively. At the same time, the standard average precision of the PointPillars detector is displayed in the unit corresponding to “No Corruption” and “standard AP”. From Table 2, we can see that existing adversarial training methods [28,29,31,33] for LiDAR scene data are effective for specific data corruption, but they will reduce the performance of the detector when exposed to other data corruptions. And adversarial training for LiDAR data slightly reduces the standard average precision of the detector in a pure environment. In addition, the adversarial purification methods DUP-Net and IF-Defense for 3D point cloud recognition significantly reduce the average precision of the target detector in all situations. Intuitively, this is because these methods focus on removing outliers and maintaining surface consistency, resulting in distant target objects in the scene being removed due to fewer points. This also proves that adversarial purification methods for 3D point cloud recognition are not suitable for LiDAR data corruption and 3D object detection models. In contrast, the proposed LiDARPure method effectively purifies scene data without targeting specific data corruption and outperforms existing adversarial training methods by more than 2% of robust AP in most cases. It should be noted that both voxel-based and pillar-based LiDARPure have almost no impact on the performance of the detector in clean scenes, maintaining the standard average precision of the detector as much as possible.

Furthermore, the LiDARPure method was also launched on PointRCNN and PV-RCNN detector. The quantitative results are presented in Table 3 and Table 4, respectively. From Table 3 and Table 4, we can see that the proposed LiDARPure method improves the average robustness precision of 3D object detectors using various point cloud representations. The higher the standard averaging precision, the higher the robust AP of the detector after it has been cleaned up from corruption by LiDARPure. This fully demonstrates the effectiveness of LiDARPure’s purification and generation strategies based on diffusion models for various types of corruption in LiDAR scene data. In addition, we can also observe that the voxel-based diffusion solution generally performs better than the pillar-based diffusion solution. This is because when the data corruption in the scene is mainly additional noise, the voxel-based solution removes the noise in each smaller unit. Adverse weather conditions essentially add a large amount of noise to the close range space of sensors, so voxel-based solutions are also effective in addressing data corruptions under adverse weather. Meanwhile, the pillar-based approach performs better in handling large-scale corruptions such as point density decrease and shear in the scene. Overall, the quantitative results in the three tables demonstrate that LiDARPure has excellent purification ability for common data corruptions in LiDAR point cloud scenes, which can significantly improve the average adversarial precision of 3D object detectors and surpass existing adversarial training strategies.

### 5.3. Visualization

In this section, we present the qualitative results of the proposed LiDARPure method. Snowfall is a representative severe weather condition, and we used the LiDARPure method to purify the LiDAR scene that was corrupted by simulated snowfall. The qualitative results are shown in Figure 2. The first line in the figure is a image captured using camera. The bird’s eye view of the scene and an enlarged view of the vehicles are shown in the second and third lines.

From the first line of Figure 2, we can see that snowfall generates a lot of noise at close range on the LiDAR and obscures the scene behind, resulting in a large number of voids in the rear space. In the corrupted scene, both the ground and target vehicles have some missing points, making it difficult to identify the points of the vehicles. After purification using LiDARPure, a large amount of noise generated by snowfall at close range has been removed from the LiDAR. And from the second line, we can learn that although it cannot recover point clouds in completely blocked void areas, more points are generated around a small number of points of the target vehicle. This greatly increases the probability of the vehicle being perceived by the detector.

Gaussian noise is a typical global data corruption in LiDAR sensor. In Figure 3, we present the qualitative results of using the LiDARPure method to purify global Gaussian noise. From the figure, we can see that although the LiDARPure method cannot fully restore the regularity of the arrangement of points in the scene and objects; it can smooth out the level of noise and make each point deviate from its original position by a smaller margin. A lower noise level makes the detection results output by the detector closer to the ground truth. In addition to improving the performance of the detector, compared to the chaotic points after corruption, the points in the purified scene appear more regular and shaped visually. However, this also indicates that diffusion models targeting each voxel or pillar unit alone are difficult to learn the arrangement properties of LiDAR point cloud data. In the process of purifying each unit, its neighborhood and global geometric information should also be considered in order to better restore the LiDAR scene.

### 5.4. Ablation Study

**Denoising recovery loss $L_{rec}$.** The denoising recovery loss $L_{rec}$ has been proposed to guide the regeneration of point clouds during the reverse diffusion process. And it mainly consists of two parts, namely diff for constraining latent features differences and pos for controlling the geometric information of point clouds. We ablate them separately to demonstrate their effectiveness. In addition, we also replaced the proposed denoising recovery loss $L_{rec}$ with traditional $l_{2}$-norm loss and Chamfer distance loss, and compared their effects on the performance of the reverse diffusion process. It is worth noting that using $l_{2}$-norm loss and Chamfer distance loss requires downsampling the number of points within the unit to be the same as before the diffusion process. The detailed comparison of the loss function is presented in Table 5 and the performance of the denoising recovery loss is based on voxel units.

It can be learnt from the Table 5 that the proposed denoising recovery loss performs better than the $l_{2}$-norm loss and Chamfer distance loss on all detectors. And both diff and pos in $L_{rec}$ help the diffusion model complete the reverse diffusion process. The performance of $L_{rec}$ may not be as good as the Chamfer distance loss when one of them is used alone, whereas the huge improvement in the mean robust AP when they are used together reflects the effectiveness of their combination.

**Impact of diffusion steps *T*.** We investigated the impact of diffusion steps *T* on the detector’s corruption robustness during the reverse diffusion process. The ablation results on the PV-RCNN detector are shown in Figure 4. From the figure, we can see that the robustness of the detector to various common data corruptions increases with *T*. Fog can cause a small number of noise points in the scene, which has a low impact on the performance of the detector. Therefore, when *T* is equal to 5, the detector also achieves high robust AP. Due to the fact that rain and snow generate a lot of noise in areas close to the sensor and block objects that are behind, its initial robust AP only exceeds 40. Because the blocked void space cannot be fully restored, its final performance is lower than 60. The global Gaussian noise and shear have a significant impact on the performance of the detector, so its robust AP is below 20 when T=5. However, as the reverse diffusion process progresses, the data corrupted by Gaussian noise can eventually be restored to a robust AP of nearly 70. And the sheared scenes can also be well restored, with a robust AP four times higher than the initial state. For adverse weather corruptions, the detector can achieve high robustness in early steps *T* and the performance stabilizes when *T* is greater than 30. However, for severe corruptions such as global Gaussian noise and object shear, the robustness average precision tends to converge only after *T* exceeds 40. This is because the noise close to LiDAR sensor generated by adverse weather can be quickly eliminated, while global point perturbation and noise require a repeated diffusion process to smooth out. Therefore, in order to make a balance between performance and computational costs, we choose T=50 to balance performance and expenses.

**Impact of unit resolution *r*.** Resolution affects the size of each unit in space and the number of its points. Due to the structural differences between voxel and pillar, it is necessary to choose different resolutions for them. We study the performance of LiDARPure at different resolutions. All types of corruption have been taken into account, so we calculated the mean of robust AP and time consumption to better select the resolution *r*. The results of voxel units and pillar units are shown in Table 6.

From the table, we can see that the optimal resolutions for voxel and pillar units are different. The excessive number or quantity of points within a unit is not conducive to the purification of diffusion processes. A pillar has an infinite vertical space with the same base area as a voxel. For a LiDAR scene with a limited number of point clouds, in order to make the number of points in the pillar and voxel units similar, it is necessary to lower the bottom resolution of the pillar. Furthermore, the resolution *r* greatly affects the time consumption of the proposed method. For both types of units, the smaller *r* leads to more units to be subjected to diffusion and denoising processes. From the table, we can understand that the single sample time consumption reaches more than 1.5 s for both units when *r* is equal to 0.1 m and the time consumption decreases as *r* increases. In order to balance the performance and time consumption, we choose r=0.2 m for voxel and r=0.15 m for pillar.

## 6. Conclusions

In this work, we propose LiDARPure, a novel approach utilizing diffusion models to effectively purify common corruptions in LiDAR data for 3D object detection. By conducting independent diffusion processes for each unit, LiDARPure overcomes the challenge of gradient confusion in adversarial training for LiDAR scenes. The utilization of the latent features of the scene as conditions for the diffusion models further improved the restoration process, resulting in a high-fidelity reconstruction of the original LiDAR scene. Our experiments have demonstrated that LiDARPure can effectively purify 19 common types of LiDAR data corruption, significantly improving the robustness of 3D object detection models against corrupted LiDAR data. In addition, the proposed method has two main limitations. First, the method fails to recover the original point cloud in a completely empty region due to severer information loss induced by LiDAR capture. Second, the computational consumption of the whole diffusion process is high, and limits its application in real-time scenarios. However, the flexibility and scalability of the proposed approach make it applicable to a wide range of LiDAR sensors and datasets. The modular design of LiDARPure allows for easy integration into existing 3D object detection pipelines, facilitating its adoption in real-world applications.

## Figures and Tables

**Figure 1 sensors-24-05440-f001:**
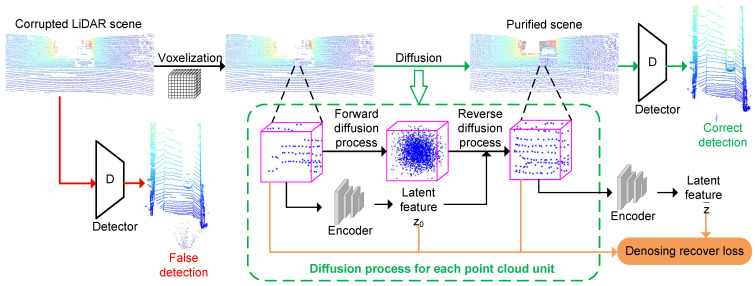
Overview of the proposed LiDARPure method for purifying the corrupted LiDAR scene. Given the corrupted LiDAR scene, it is divided into independent point cloud units through voxelization. LiDARPure utilizes a diffusion model to purify the point cloud within each unit. The corrupted points within the unit are converted into noise through a forward diffusion process, and then restored to purified point clouds through reverse diffusion process. The latent feature of the point cloud within the unit are extracted using the encoder and participate as the condition in the reverse diffusion process. Each unit and its latent features before ($z_{0}$) and after ($\bar{z}$) purification are used to calculate the denoising recovery loss. All units complete the diffusion process to form a purified scene.

**Figure 2 sensors-24-05440-f002:**
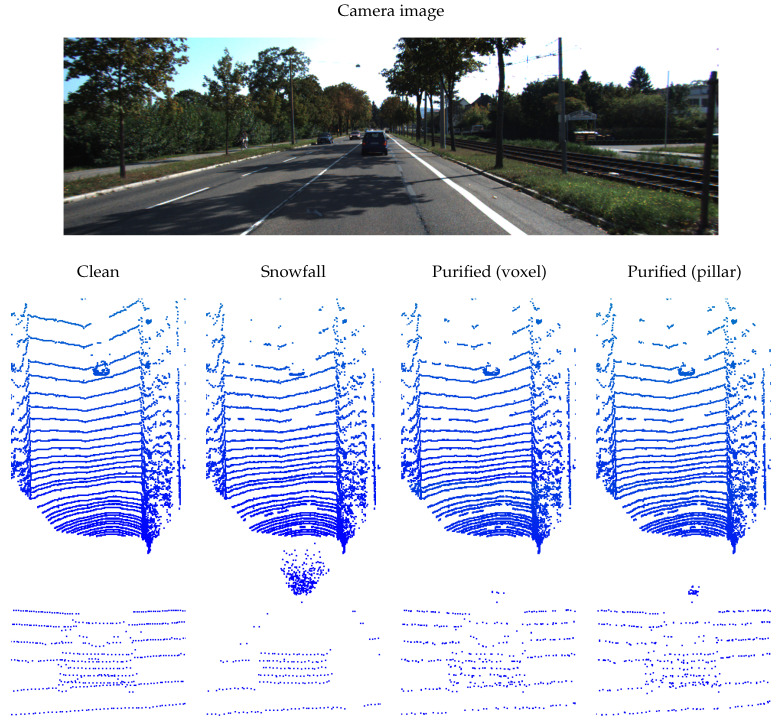
The qualitative results of the proposed LiDARPure method purify the LiDAR scene corrupted by snowfall. The units used are indicated in the brackets.

**Figure 3 sensors-24-05440-f003:**
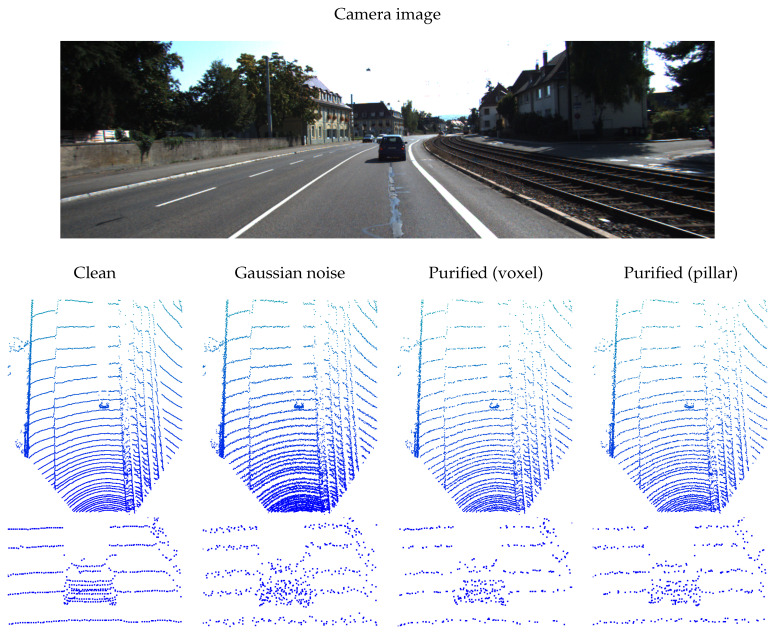
The qualitative results of proposed LiDARPure method purifies the LiDAR scene corrupted by Gaussian noise. The units used are indicated in bracket.

**Figure 4 sensors-24-05440-f004:**
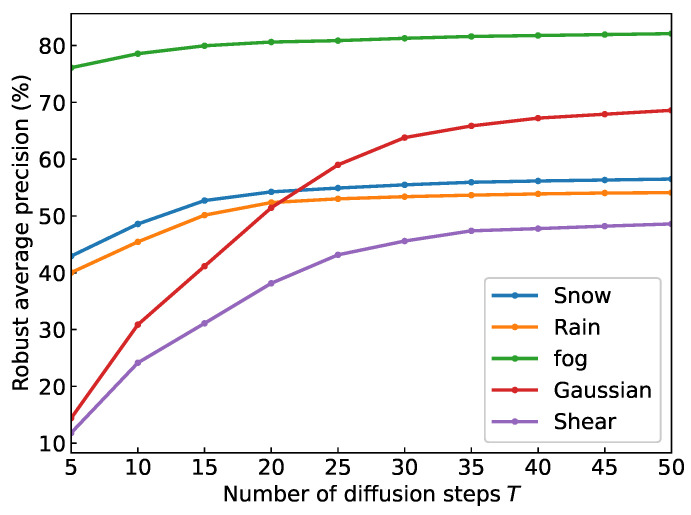
Robust AP affected by different steps *T* on PV-RCNN detector in the reverse diffusion process.

**Table 1 sensors-24-05440-t001:** Average precision (AP) of PointPillars detector with different adversarial training strategies when experiencing data corruptions.

Corruptions	Adversarial Training			
	None	Snow	Rain	Fog
None	**78.4**	78.1	77.5	77.2
Snow	36.5	39.5	34.0	33.1
Rain	36.2	35.2	38.8	35.7
Fog	64.3	64.1	63.8	66.3

**Table 2 sensors-24-05440-t002:** The standard AP (%) and robust AP (%) of various defense strategies when exposed to 19 types of data corruption on PointPillars detector. The basic unit for diffusion process is represented in bracket. The best results are shown in bold and previous best results are underlined.

		Defense Strategies (Robust AP)							
Corruptions	(Standard AP)	Snowfall Simulation [31]	LISA [28]	Fog Simulation [29]	Teufel et al. [33]	DUP-Net [34]	IF-Defense [37]	Ours (Voxel)	Ours (Pillar)
No Corruption	78.4	78.1	77.5	77.9	78.0	54.2	59.4	78.3	**78.4**
Snow	36.5	39.5	34.0	33.1	37.8	26.2	28.2	**44.6**	44.0
Rain	36.2	35.2	38.8	35.7	39.6	25.3	26.2	**43.9**	43.1
Fog	64.3	64.1	63.8	66.3	64.3	43.4	47.4	**71.7**	71.5
Sunlight	62.3	61.4	61.7	61.5	61.7	45.0	48.7	**66.8**	66.4
Density	76.5	76.2	75.9	76.2	75.4	54.0	56.9	77.2	**77.6**
Cutout	70.3	70.1	70.0	70.4	69.8	50.0	54.1	73.0	**73.2**
Crosstalk	70.9	68.7	69.3	68.8	69.0	48.9	50.5	71.6	**72.5**
Gaussian	74.7	74.0	73.8	74.2	73.8	50.4	59.3	**77.1**	76.5
Uniform	77.3	77.2	76.8	77.2	76.4	56.3	56.1	**78.0**	77.7
Impulse	78.2	78.2	78.1	78.3	78.1	54.1	60.9	78.2	78.2
Moving object	50.2	50.3	50.0	52.1	52.5	33.8	37.4	48.4	50.2
Local Density	69.6	69.4	69.5	69.7	69.9	48.8	52.4	**73.6**	73.4
Local Cutout	61.8	60.6	61.6	60.5	60.4	41.9	44.3	**66.2**	65.7
Local Gaussian	76.7	76.7	76.4	77.0	76.5	52.8	60.7	**77.7**	77.6
Local Uniform	78.0	78.0	77.9	78.2	78.0	54.2	59.5	**78.2**	77.9
Local Impulse	78.4	78.4	78.3	78.4	78.4	53.3	56.5	**78.5**	78.4
Shear	39.6	36.9	39.0	37.8	36.6	28.7	31.2	39.9	**40.4**
Scale	70.3	70.2	69.9	70.6	69.4	47.7	51.4	**73.3**	73.2
Rotation	72.7	72.8	72.3	72.7	72.6	50.5	57.9	**73.4**	72.2
Average	65.5	65.2	65.1	65.2	65.3	45.5	49.5	**68.0**	67.9

**Table 3 sensors-24-05440-t003:** The standard AP (%) and robust AP (%) of various defense strategies when exposed to 19 types of data corruption on PointRCNN detector. The basic unit for diffusion process is represented in bracket. The best results are shown in bold and previous best results are underlined.

		Defense Strategies (Robust AP)							
Corruptions	(Standard AP)	Snowfall Simulation [31]	LISA [28]	Fog Simulation [29]	Teufel et al. [33]	DUP-Net [34]	IF-Defense [37]	Ours (Voxel)	Ours (Pillar)
No Corruption	80.6	80.5	80.1	79.4	80.2	55.9	58.9	80.3	80.0
Snow	50.4	54.3	49.8	49.9	52.7	36.0	36.3	**55.9**	54.5
Rain	51.3	51.0	53.4	50.6	51.9	36.4	36.8	**54.2**	53.8
Fog	72.1	71.6	71.9	72.7	72.8	51.6	54.7	**73.7**	73.1
Sunlight	62.8	61.6	62.1	62.0	62.2	42.5	48.3	**64.3**	63.6
Density	80.4	78.2	80.5	79.7	79.4	56.0	62.3	80.3	**80.5**
Cutout	73.9	73.6	71.6	73.2	72.5	52.1	53.4	74.7	**75.1**
Crosstalk	71.5	69.8	71.2	71.5	71.3	51.6	52.0	73.5	**73.7**
Gaussian	61.2	60.9	60.0	60.1	60.3	42.5	47.6	**64.1**	63.6
Uniform	76.4	76.5	74.9	76.3	76.6	52.1	55.5	**77.5**	77.2
Impulse	79.8	78.9	78.8	77.8	79.1	56.9	57.0	**80.2**	80.0
Moving object	50.5	50.2	50.5	50.3	50.2	34.7	37.8	51.9	**52.6**
Local Density	74.2	72.8	72.7	73.1	73.2	52.4	54.9	**77.7**	76.8
Local Cutout	67.9	66.2	66.6	67.4	66.7	45.8	48.9	**69.9**	69.7
Local Gaussian	69.8	68.7	67.5	69.2	69.0	47.5	54.3	**72.4**	72.1
Local Uniform	77.7	77.5	76.1	76.5	77.4	53.9	58.9	**79.6**	79.3
Local Impulse	80.3	80.1	78.9	79.7	79.8	54.6	61.5	80.2	**80.3**
Shear	39.8	39.7	39.8	39.1	39.5	27.0	30.7	40.9	**41.4**
Scale	71.5	71.6	70.3	71.3	71.1	49.8	54.5	**74.1**	73.5
Rotation	75.6	75.4	75.4	74.4	75.3	51.6	54.6	**77.2**	77.0
Average	67.7	67.3	66.9	67.1	67.4	47.1	50.5	**69.6**	69.3

**Table 4 sensors-24-05440-t004:** The standard AP (%) and robust AP (%) of various defense strategies when exposed to 19 types of data corruption on PV-RCNN detector. The basic unit for the diffusion process is represented in the brackets. The best results are shown in bold and previous best results are underlined.

		Defense Strategies (Robust AP)							
Corruptions	(Standard AP)	Snowfall Simulation [31]	LISA [28]	Fog Simulation [29]	Teufel et al. [33]	DUP-Net [34]	IF-Defense [37]	Ours (Voxel)	Ours (Pillar)
No Corruption	84.4	84.0	83.9	83.8	84.0	58.0	65.5	84.2	84.1
Snow	52.4	55.7	51.6	51.2	54.6	37.7	39.2	**56.5**	56.0
Rain	51.6	51.3	53.1	50.8	52.8	36.0	39.6	**54.1**	53.7
Fog	79.5	79.0	79.2	80.7	80.4	54.7	57.8	**82.1**	81.6
Sunlight	79.9	79.4	78.2	79.1	79.7	53.6	60.6	**82.0**	81.3
Density	82.8	82.6	81.0	79.5	81.7	57.8	63.1	83.2	**83.5**
Cutout	76.1	74.3	74.3	76.2	75.6	53.1	55.3	77.5	**78.0**
Crosstalk	82.3	80.4	79.6	82.4	82.6	56.9	58.7	82.6	**82.8**
Gaussian	65.1	63.4	63.7	63.0	64.0	43.9	50.1	**68.6**	68.4
Uniform	81.2	80.6	79.2	78.4	80.4	58.4	62.8	82.1	**82.2**
Impulse	82.8	81.6	81.4	80.4	81.1	57.4	62.1	**83.3**	83.0
Moving object	54.6	53.7	53.0	53.4	54.0	39.0	42.1	**55.8**	55.5
Local Density	77.6	75.3	76.6	75.6	76.9	56.2	55.2	**80.3**	79.8
Local Cutout	72.3	70.6	70.7	71.3	71.4	50.4	51.9	75.8	**75.9**
Local Gaussian	70.4	68.5	69.5	67.9	69.4	50.3	51.4	**74.1**	73.4
Local Uniform	82.1	80.3	80.9	79.4	80.5	58.0	63.6	**83.4**	83.2
Local Impulse	84.0	81.7	81.1	83.2	82.6	58.7	61.0	84.0	**84.1**
Shear	47.7	47.0	46.7	47.4	47.0	33.0	36.2	48.6	**49.3**
Scale	76.8	75.0	75.9	74.3	75.7	54.8	55.0	78.7	**78.8**
Rotation	79.9	77.6	77.9	79.7	79.0	55.1	62.0	**80.4**	80.0
Average	72.6	71.5	71.2	71.3	72.1	50.8	54.1	**74.4**	74.2

**Table 5 sensors-24-05440-t005:** The mean robust AP (%) of all data corruptions under different loss function on three victim detectors.

Denoising Recovery Loss		*l_2_-norm*	Chamfer Distance	Mean Robust AP		
*diff*	*pos*			PointPillars	PointRCNN	PV-RCNN
-	-	✓	×	64.1	67.4	69.7
-	-	×	✓	65.9	68.3	71.3
✓	×	-	-	65.6	68.8	72.8
×	✓	-	-	66.4	68.2	71.5
✓	✓	-	-	68.0	69.6	74.4

**Table 6 sensors-24-05440-t006:** The mean robust AP (%) of all data corruptions and time consumption (s) under different resolutions *r*.

	*r* for Voxel					*r* for Pillar				
	0.1	0.15	0.2	0.25	0.3	0.1	0.15	0.2	0.25	0.3
PointPillars	64.1	67.8	68	65.2	58.9	68.1	67.9	64.2	60.6	52.3
PointRCNN	64.0	68.3	69.6	65.0	59.7	67.8	69.3	67.8	62.0	57.1
PV-RCNN	67.4	72.6	74.4	69.2	62.2	72.9	74.2	72.0	69.5	64.8
time consumption	2.74	2.16	1.31	0.85	0.64	1.67	1.29	0.94	0.56	0.33

## Data Availability

The raw data supporting the conclusions of this article will be made available by the authors on request.

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
