# Peer review of "Diffusion Models-Based Purification for Common Corruptions on Robust 3D Object Detection"

_sensors, 2024, doi:10.3390/s24165440_

Round 1

Reviewer 1 Report

Comments and Suggestions for Authors

This study leverages the powerful generation ability of diffusion models to purify corruption in the LiDAR scene data. The problems and methods are clearly presented, however, in my opinion, the effectiveness and the advantages of the proposed method are not well demonstrated. My major concerns are as followed.

1. The comparison algorithms are not sufficient. It is recommended to compare with the latest SOTA method published in 2023 or 2024. The most recent comparison algorithm in the comparative experiment was in 2021, and three years have passed.

2. In the second part, Related Work”, there is a lack of literature review on the core issue of this manuscript , and 3D detection or the Corruption are not the focus of this study, which feels to avoid the heavy points and touch on the light ones.

3.I didn't quite understand the ablation experiment, it looks more like a parameter optimization experiment.

4. The author did not provide the time consumption of the method, which appears to be a relatively refined purification process. What is the practical significance of its application?

And can the improvement of the detector compensate for the problem of data purification?

5. The author claimed that the loss function is their contribution, but no experimental verification has been conducted on this contribution in the experiment part.

6. What are the limitations of the proposed method? The authors could provide these in the conclusion section.

Reviewer 2 Report

Comments and Suggestions for Authors

Dear Sensors assistant editor Pimpakan Suwanich,

After carefully reading the manuscript #3147056: “Diffusion models based purification for common corruptions on robust 3D object detection” I consider that it can be published in Sensors provided a minor revision is performed.

The manuscript proposes a method to purify common corruptions in large-scale Light Detection and Ranging (LiDAR) scene data.  The method is compared against adversarial training and shows a better performance with the possibility of handling various types of data corruption without being hindered by data sparsity and gradient confusion.

The manuscript is clearly written and the state of the art is also clearly established. However, I would like the authors to address the following issues before publication:

1)     The Intro is clearly written but it lacks of a basic scheme of information capture through LiDAR.

2)     The authors report 19 common corruptions in LiDAR data but there are almost no images of the scenes under study (only point cloud results which might be difficult to understand are included in the manuscript).

3) denoising recovery loss term considers additive Gaussian noise as the model for data corruption: is there any other, non-Gaussian source of noise in the system? Might the sensor introduce some shot-noise?

Round 2

Reviewer 1 Report

Comments and Suggestions for Authors

none